



# Strategies for Conservative and Non-Conservative Monotone Remapping on the Sphere

David Marsico[1] and Paul Ullrich[2]

[1]Department of Mathematics, University of California-Davis, Davis, CA 95616
[2]Department of Land, Air and Water Resources, University of California-Davis, Davis, CA 95616

**Correspondence:** David Marsico (dhmarsico@ucdavis.edu)

**Abstract.** Monotonicity is an important property of remapping operators for coupled weather and climate models. However, it is often challenging to design highly accurate operators that avoid the generation of new extrema or keep a remapped field between physically prescribed bounds. To that end, this paper explores several traditional and novel approaches for both conservative and non-conservative monotone remapping on the sphere. The accuracy and effectiveness of these algorithms are
evaluated in the context of several different real and idealized fields and meshes.

## 1   Introduction

An important operation in global climate models is the transferring, or remapping, of data between different component grids. For example, information needs to be exchanged at the interface between the atmosphere and ocean models, when both are typically defined on different grids. Atmospheric models often use icosahedral or cubed sphere grids, while ocean models have

relied on unstructured meshes (Satoh et al., 2008; Taylor et al., 2007; Ringler et al., 2013). Remapping of data between grids whose structures differ greatly is a challenging and important problem, as doing so inaccurately can impact the stability of coupled simulations (Beljaars et al., 2017). There are other circumstances where accurate remapping operators are important, such as post-processing and mesh refinement. In the former case, the grid on which a simulation is performed may not be ideal for carrying out analysis, and transferring data onto a structured mesh that is more amenable for analysis is often useful.

In the latter case, grid nesting (Harris and Lin, 2014) and adaptive grids (Jablonowski et al., 2006; Skamarock and Klemp, 1993) have been used to resolve the complex multiscale nature of the atmosphere. Ensuring the accurate interpolation of data between the different component grids in these simulations is crucial to preserving the models overall accuracy (Slingo et al., 2009; Mahadevan et al., 2020).

There are a number of desirable properties of remapping operators, in addition to accuracy. These properties include consis-

tency, conservation, and monotonicity and correspond, respectively, to the mapping of the constant field to the constant field, preservation of total mass, and no generation of new extrema (Ullrich and Taylor, 2015). These properties are necessary for ensuring important physical consistency of model simulations. Some fields, like mass (which is usually stored as density), are required to be conserved, while others, like tracers or mixing ratios, are required to satisfy certain bounds following the remap-





ping process. It is therefore imperative for schemes that remap these fields to preserve conservation and global monotonicity
constraints, so as not to introduce artificial sources of error (Kritsikis et al., 2017).

The main property of remapping schemes that we are concerned with in this paper is monotonicity. In the case of conservative remapping, monotonicity is often achieved by way of limiters (Barth and Jespersen, 1989). In the conservative case, we are interested in applying the "clip and assured sum" (CAAS) method, which acts as a post-processing filter operation on the remapped field (Bradley et al., 2019). In the non-conservative case, we are interested in linear monotone remapping operators

that depend only on the mesh structures, and can be computed once and then applied in an offline manner.

TempestRemap (Ullrich and Taylor, 2015; Ullrich et al., 2016) is a widely used package for generating conservative, consistent and (optionally) monotone linear maps between arbitrary grids on the sphere, with data stored as volume averages (finite volume method) or as coefficients of a finite element expansion. Although conservative remapping is necessary for ensuring, for example, fluxes at the ocean-atmosphere interface preserve global invariants, when remapping states or vectors it is often

the case that monotonicity and accuracy are more important than conservation. Indeed, when mapping from a coarse grid to a fine grid, conservative and monotone schemes will appear blocky because each fine grid volume completely within a coarse grid volume must exactly preserve the state of the coarse grid volume. Consequently, the Energy Exascale Earth System Model (Golaz et al., 2019, E3SM), which uses TempestRemap maps under the hood, falls back on bilinear maps for transferring state data when mapping from coarse resolution to fine resolution grids. To support the operational remapping of data in this manner,

the methods developed in this paper have been implemented in v2.1.6 of the TempestRemap package (Ullrich et al., 2022).

This paper consists of three main sections. First, we will describe the basic setup of remapping problems, the test cases that are used in our numerical experiments, and the metrics used to asses the accuracy of the remapping schemes. In the next section, we will look at monotone conservative remapping. In general, it is difficult to construct remapping operators that satisfy conservation and monotonicity, while still maintaining high order accuracy. So one of the main purposes of this section

is to examine the extent to which a conservative and monotone remapping operator can maintain the accuracy of it's non-monotone counterpart. We will also analyze the effectiveness of this conservative and monotone operator in minimizing the errors associated with the remapping of discontinuous source fields, as well its ability to remap real data fields accurately. The subject of the next section is non-conservative monotone remapping, and it is divided into two main parts. The first part focuses on traditional approaches to monotone remapping, and includes a description of the bilinear method used in the Earth System

Modeling Framework (ESMF) (Hill et al., 2004), as well as two additional approaches that may provide advantages in some circumstances. In the second part we, we show that the accuracy of these traditional approaches is reduced when remapping from source meshes that are finer than the target mesh, and a method is introduced to correct this. We end with conclusions and future research directions.





## 2 Preliminaries

Let $\Omega$ denote the unit sphere. Given a source mesh, $\Omega^s$, and a target mesh, $\Omega^t$, the remapping operator, $\mathbf{R}$, is a matrix constructed to satisfy

$$\boldsymbol{\psi}^t = \mathbf{R}\boldsymbol{\psi}^s, \tag{1}$$

where $\boldsymbol{\psi}^s = (\psi_1^s, \ldots, \psi_s^{f_s}) \in \mathbb{R}^{f_s}$, and $\boldsymbol{\psi}^t = (\psi_1^t, \ldots, \psi_t^{f_t}) \in \mathbb{R}^{f_t}$, are vectors of discrete density values on the source and target meshes, respectively. The number of degrees of freedom on the source mesh is denoted by $f_s$, and the number of degrees of

freedom on the target mesh is denoted by $f_t$. Here, $\boldsymbol{\psi}^s$ corresponds to the discretization of a function $\psi : \Omega^s \to \mathbb{R}$, either by sampling $\psi$ at a a set of discrete nodes by pointwise sampling, or over a set of regions, by area averaging. The operators that discretize the function $\psi$ into the discrete vectors $\boldsymbol{\psi}^s$ and $\boldsymbol{\psi}^t$ are denoted by $\boldsymbol{D}^s$ and $\boldsymbol{D}^t$. Degrees of freedom on the source and target meshes can be stored in various ways, though in this paper we focus on finite-volume or finite-element methods. In the former case, degrees of freedom on the mesh correspond to area or volume averages, and in the latter, they are stored

as coefficients of basis functions with compact support. For instance, for the spectral element method, a type of finite element method, it is typical to store degrees of freedom at a set $N_p^2$ Gauss-Lobatto-Legendre (GLL) nodes within each face.

Following Ullrich and Taylor (2015), the metrics that are used to asses the accuracy of the remapping schemes in this paper are as follows:

$$L_1 \equiv \frac{I^t[|\mathbf{R}\mathbf{D}^s(\psi) - \mathbf{D}^t(\psi)|]}{I^t[|\mathbf{D}^t(\psi)|]} \tag{2}$$

$$L_2 \equiv \frac{\sqrt{I^t[|\mathbf{R}\mathbf{D}^s(\psi) - \mathbf{D}^t(\psi)|^2]}}{\sqrt{I^t[|\mathbf{D}^t(\psi)|^2]}} \tag{3}$$

$$L_\infty \equiv \frac{\max|\mathbf{R}\mathbf{D}^s(\psi) - \mathbf{D}^t(\psi)|}{\max|\mathbf{D}^t(\psi)|} \tag{4}$$

$$L_{\min} \equiv \frac{\min|\mathbf{R}\mathbf{D}^s(\psi)| - \min|\mathbf{D}^t(\psi)|}{\min|\mathbf{D}^t(\psi)|} \tag{5}$$

$$L_{\max} \equiv \frac{\max|\mathbf{R}\mathbf{D}^s(\psi)| - \max|\mathbf{D}^t(\psi)|}{\max|\mathbf{D}^t(\psi)|}, \tag{6}$$

where $I^s$ and $I^t$ are the integration operators on the source mesh given by

$$I^s[\boldsymbol{\psi}^s] = \sum_{i=1}^{f^s} \psi_i^s J_i^s, \tag{7}$$





with $J_i^s$ denoting the weight of the $i$th degree of freedom on the source mesh. The integration operator on the target mesh, $I^t$, is defined similarly.

We will use several idealized test cases for our numerical experiments, including a low frequency harmonic denoted by $Y_2^2$, and given by the equation

$$\psi = 2 + \cos^2(\theta)\cos(\lambda), \tag{8}$$

a high frequency harmonic, $Y_{32}^{16}$, given by

$$\psi = 2 + \sin^{16}(2\theta)\cos(16\lambda) \tag{9}$$

and a vortex represented by

$$\psi = 1 - \tanh\left[\frac{\rho'}{d'}\sin(\lambda' - \omega' t)\right], \tag{10}$$

where $r' = r_0\cos(\theta')$ is the radius, $\omega$ is the angular velocity with

$$\omega = \begin{cases} 0, & \text{if } \rho' = 0 \\ \dfrac{V_t}{\rho'} & \text{if } \rho' \neq 0, \end{cases} \tag{11}$$

and $V_t$ is the tangential velocity with

$$V_t = \frac{3\sqrt{3}}{2}\text{sech}^2(\rho')\tanh(\rho'). \tag{12}$$

Here, $(\lambda', \theta')$ are the coordinates in a rotated spherical coordinate system whose pole is at $(0, 0.6)$, and we set $r_0 = 3$, $d = 5$, and $t = 6$ (Ullrich and Taylor, 2015).

## 3 Monotone Conservative Remapping

The focus of this section is on monotone conservative remapping, and assessing potential improvements in accuracy that arise from employing a nonlinear remapping technique to enforce bounds preservation. We consider fields whose total mass needs to be conserved across the remapping process, and that need to remain between specified bounds. This form of "bounds preservation" is important for fields such as mixing ratios, which are required to remain between zero and unity, and it corresponds to a global form of monotonicity where no new global extrema are generated. We also consider local forms of bounds preservation, which are stronger than global monotonicity in the sense that they will not introduce any new local extrema.

High order remapping methods can lead to overshoots or undershoots of the remapped field, which is problematic for several reasons. For instance, high order remapping of discontinuous source fields may lead to oscillatory behavior of the remapped field similar to the Gibbs phenomenon (Gottlieb and Shu, 1997; Mahadevan et al., 2022). Preserving the bounds of these fields, as well as minimizing Gibbs oscillation is critical to maintaining the accuracy of coupled simulations.





One method used to guarantee bounds preservation is the "Clip and Assured Sum" (CAAS) method (Bradley et al., 2019), whereby the remapped field is cropped, and then mass is redistributed in such a way that the field remains within specified bounds. Specifically, given a vector of source values, and lower and upper bounds $l$ and $u$, the CAAS algorithm modifies the remapped field, $\mathbf{R}\psi^s$, such that $l \leq \mathbf{R}\psi^s \leq u$ while still preserving conservation. In this section, our goal is to examine the utility of the CAAS algorithm as a way of ensuring bounds preservation and reducing Gibbs phenomena while still ensuring accuracy, and conservation. In particular, we are interested in documenting the effect of CAAS on standard error norms, as implemented in TempestRemap.

### 3.1 Finite-Volume to Finite-Volume

Here, we look at the case where the source and target meshes are both finite-volume. In particular, we are interested in applying the CAAS algorithm with two different types of local bounds preservation, which we now describe.

We let

$$a_i = \min_{\text{intersecting faces}} \boldsymbol{\psi}_i^s, \qquad b_i = \max_{\text{intersecting faces}} \boldsymbol{\psi}_i^s \tag{13}$$

where the maximum and minimum values are computed over all source faces that intersect target face $i$. We then define the local lower and upper bounds, $\boldsymbol{l}_{l,i}$, and $\boldsymbol{u}_{l,i}$, as

$$\boldsymbol{l}_{l,i} = a_i, \qquad \boldsymbol{u}_{l,i} = b_i. \tag{14}$$

We have found that a variation of this type of bounds preservation gives an improvement in convergence under mesh refinement, which we call "local-p bounds preservation," and describe as follows. We define the minimum and maximum value over a set of adjacent faces as

$$c_i = \min_{\text{adjacent faces}} \boldsymbol{\psi}_i^s, \qquad d_i = \max_{\text{adjacent faces}} \boldsymbol{\psi}_i^s \tag{15}$$

Here, the minimum and maximum are computed over a set of $(p+1)^2$ source faces adjacent to target face $i$, where $p$ is the order of the polynomial reconstruction. The choice of $(p+1)^2$ was an empirical one that provided good convergence results. The local-p lower and upper bounds, $\mathbf{l}_{p,i}$, and $\mathbf{u}_{p,i}$, as are then defined as

$$\boldsymbol{l}_{p,i} = \min(a_i, c_i), \qquad \boldsymbol{u}_{p,i} = \max(b_i, d_i). \tag{16}$$

For our numerical tests, we use cubed sphere source meshes with $n_e \times n_e$ elements per panel, for $n_e = 15, 30, 60$. The target is a 1° latitude-longitude mesh with 360 longitudinal elements and 180 latitudinal elements. The convergence results for remapping with and without CAAS with local-p bounds preservation for several different fields are presented in figures 1 and 2. For each mesh, we plot the errors as functions of $np$, the order of the polynomial reconstruction on the source mesh.





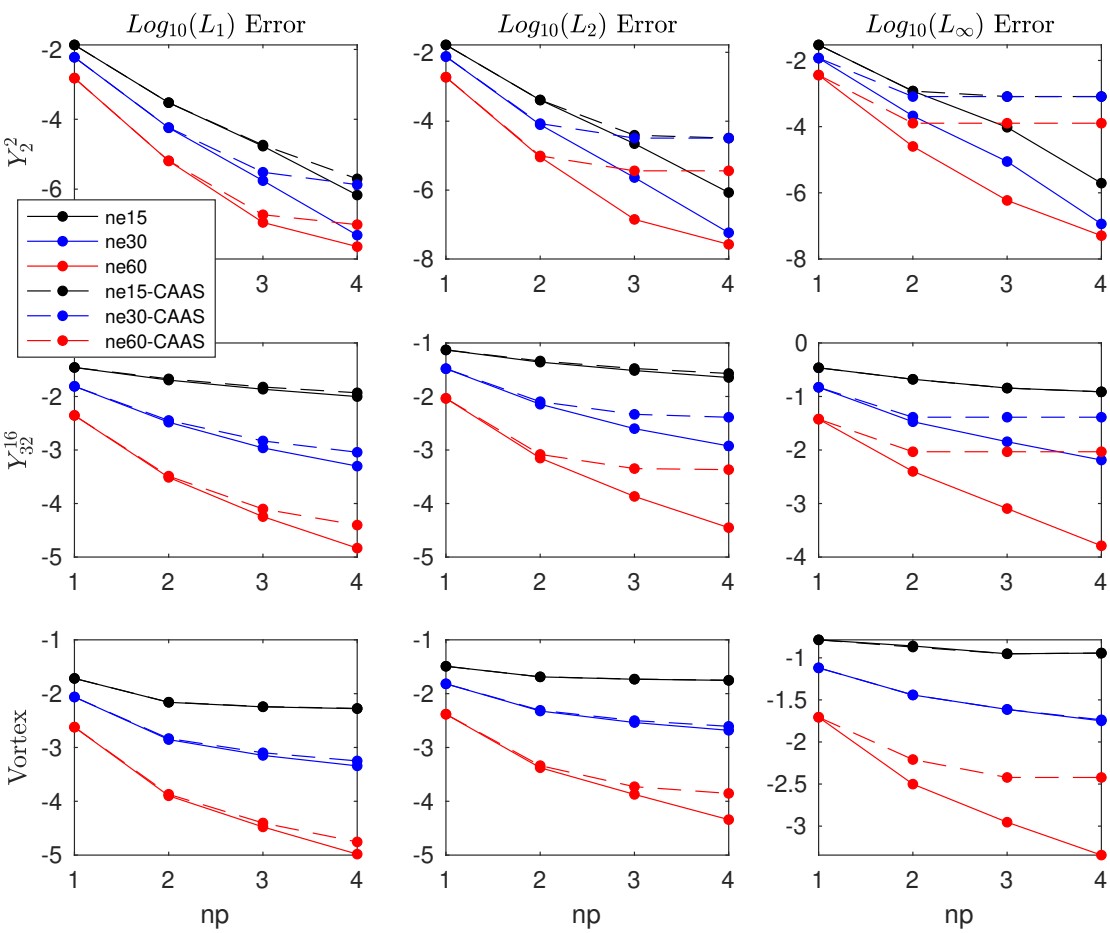

**Figure 1.** Convergence test for finite-volume to finite-volume remapping from cubed spheres to a $1°$ latitude-longitude mesh for three different test cases. The dashed lines show the results using CAAS with local-p bounds preservation, and the solid lines are the results without CAAS.

In all cases, the $L_1$ convergence for the remapped field both with and without CAAS are very similar, and the $L_2$ convergence

is qualitatively similar as well. However, the $L_\infty$ error levels off for all three test cases when CAAS is applied, particularly for the high resolution cases. This can be understood by looking at the convergence of the corresponding $L_{min}$ and $L_{max}$ errors. For all test cases, the remapped field overshoots and undershoots the global maximum and minimum of $\mathbf{D}^t(\psi)$, that is, $\psi$ evaluated on the target mesh. The CAAS algorithm will then clip these fields so that, at the very least, they respect the global bounds of $\psi$ evaluated on the source mesh, $\mathbf{D}^s(\psi)$. In this case, the remapped field, after applying CAAS to it, will satisfy the inequality

$\min \mathbf{D}^s(\psi) \leq \mathbf{R}\boldsymbol{\psi}^s \leq \max \mathbf{D}^s(\psi)$. The effect of the clipping operation, then, is that the $L_\infty$ error will be approximately as large

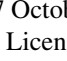


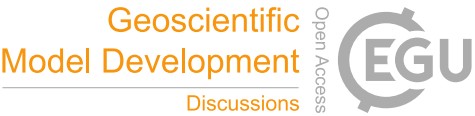

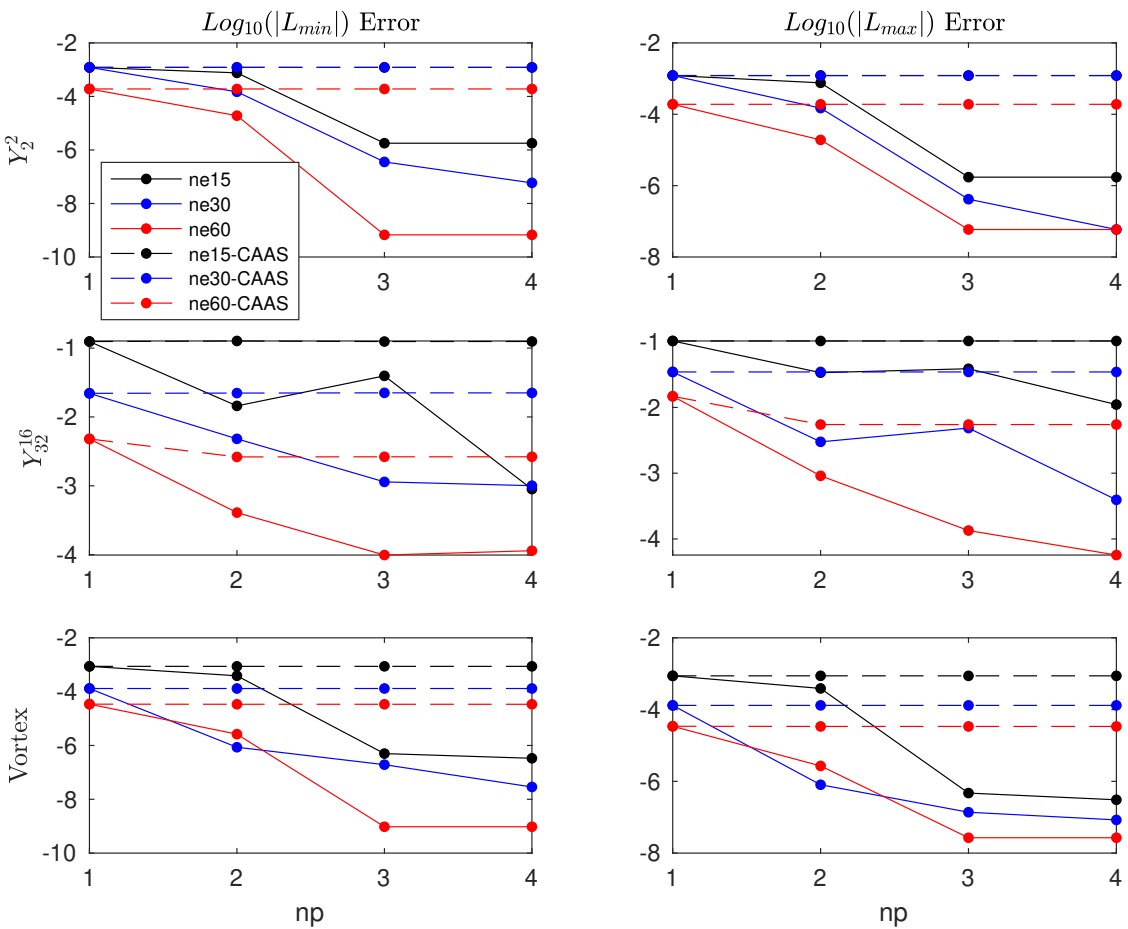

**Figure 2.** Convergence test for finite-volume to finite-volume remapping from a cubed sphere to a $1°$ latitude-longitude mesh for three different test cases. Note that in all cases for np > 1, the remapped field overshoots and undershoots the absolute maximum and minimum, respectively.

as

$$\min(|\max \mathbf{D}^t(\psi) - \max \mathbf{D}^s(\psi)|, |\min \mathbf{D}^t(\psi) - \min \mathbf{D}^s(\psi)|), \qquad (17)$$

because the minimum and maximum of the remapped field after applying CAAS to it will be approximately equal to $\min \mathbf{D}^t(\psi)$, and $\max \mathbf{D}^t(\psi)$. As can be seen in figure 2, the $L_{min}$ and $L_{max}$ errors remain essentially constant for all mesh resolutions as the

140 order of the polynomial reconstruction is increased. This constancy then results in an effective lower bound on the $L_\infty$ errors, and is the reason for the flat-lines for the $Y_2^2$, $Y_{32}^{16}$, and vortex test cases.





## 3.2 Finite Element to Finite Volume

Here, we examine bounds preservation in the case where the source mesh is finite element. Local bounds preservation is defined similarly to how it was for finite volume source meshes, but now the minimum and maximum in equation (14) are computed over all GLL nodes on all the faces that intersect target face $i$. The convergence results for standard remapping with and without CAAS with local bounds preservation for several different fields are presented in figures 3 and 4. Here, the convergence results are nearly identical, apart from the $L_\infty$ error for fourth order reconstruction for the $Y_2^2$ test case on the coarsest mesh. By looking at the corresponding $L_{\min}$ and $L_{\max}$ errors, we see that the remapped field undershoots and overshoots the global minimum and maximum of $\mathbf{D}^t(\psi)$. The error induced by CAAS will once again be approximately equal to the expression given in equation (17).

## 3.3 Gibbs Phenomenon

In this section, we examine the effectiveness of CAAS in reducing overshoots and undershoots associated with remapping a discontinuous source field. To that end, we modify the vortex test case by defined by equation (10) by letting the field be equal to zero if it is less than a certain threshold, and equal to one if it is greater than it.

In figure 5, results are shown for four different schemes applied to the vortex test: remapping without CAAS, CAAS with local bounds preservation, local-p bounds preservation, and global bounds preservation. By global bounds preservation, we mean that the remapped field satisfies the equation

$$\min \mathbf{D}^s(\psi) \leq \mathbf{R}\psi^s \leq \max \mathbf{D}^s(\psi). \tag{18}$$

It is evident from this figure that without applying CAAS, the remapped field suffers from a significant loss of accuracy close to the discontinuity, and significant undershoots of the global minimum are present. The fields obtained using CAAS with local bounds, local-p bounds, and global bounds preservation all show a reduction of oscillations and an improvement of accuracy, with the local bounds preservation resulting in the greatest improvement. One-dimensional cross-sections of the remapped fields allow us to examine this reduction more closely. In particular, observe from figure 6 how sharply CAAS with local bounds represents the discontinuity. Although the remapped fields obtained using CAAS with local-p bounds and global bounds remain between zero and unity, there are still slight oscillations near zero.

## 3.4 Real Data

To test the performance of the CAAS algorithm on real data, we use the cloud fraction data generated from the MIRA real data emulator (Mahadevan et al., 2022), which is a field that is required to be bounded between zero and unity, and is shown in figure 7. We perform two tests. The first is remapping the cloud fraction from an ne90 cubed sphere mesh to a $1°$ latitude-longitude mesh, and the second is from an ne360 cubed sphere to a $0.25°$ degree latitude-longitude mesh. In each case, we compare the accuracy of first order, second order, and second order using CAAS with global bounds preservation between zero and unity. We see from tables 1 and 2 that the CAAS algorithm gives the smallest error norms for all metrics in both tests.



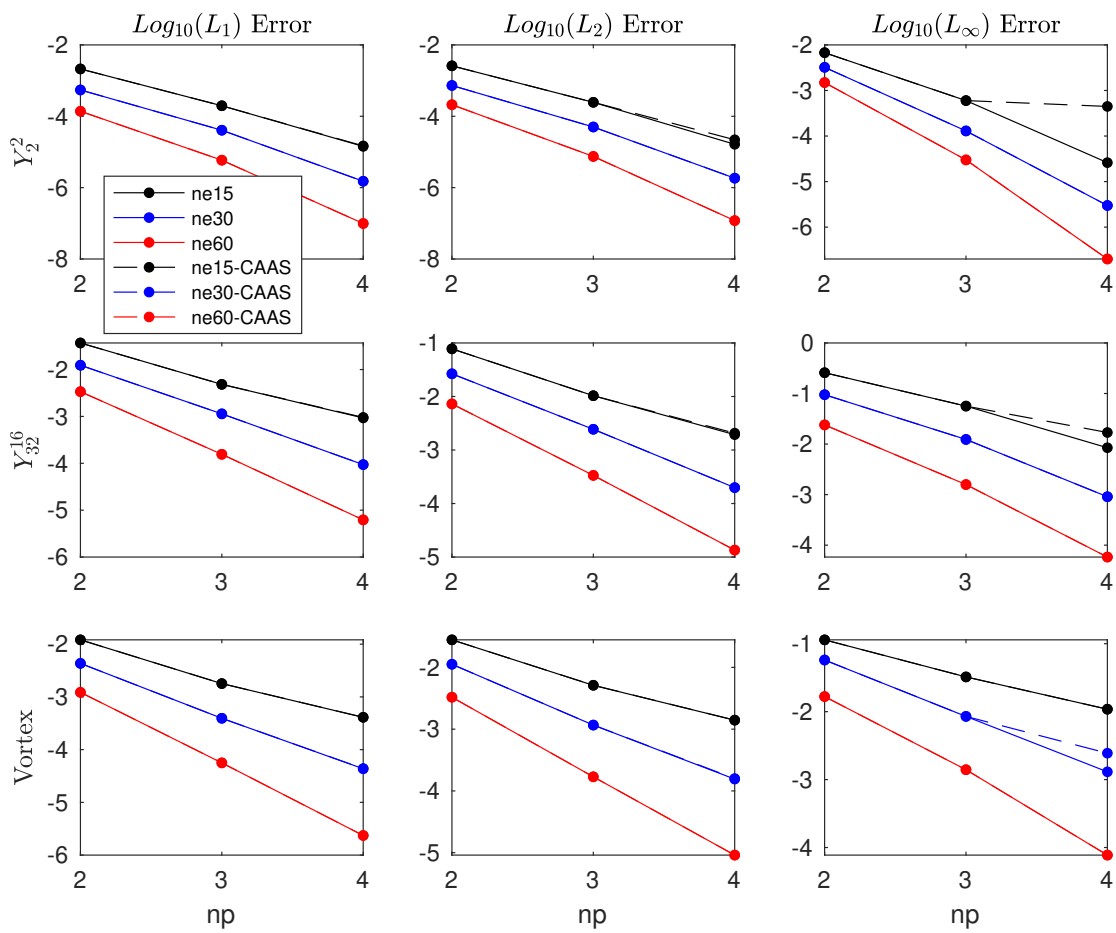

**Figure 3.** Convergence test for finite-element to finite-volume remapping from a cubed sphere to latitude-longitude mesh using local bounds preservation for three different test cases. The setup is the same as it was for finite-volume to finite-volume remapping, with the dashed lines showing the results using CAAS with local bounds preservation, and the solid lines showing the results without CAAS.

**Table 1.** Error Norms for remapping from an ne90 cubed sphere to a $1°$ degree latitude-longitude mesh.

| Method | $L_1$ | $L_2$ | $L_\infty$ | $L_{min}$ | $L_{max}$ |
|---|---|---|---|---|---|
| 1st order | $8.69574e-03$ | $1.0483e-02$ | $6.78599e-02$ | $-1.60710e-03$ | $0.0$ |
| 2nd order | $6.96131e-03$ | $7.36387e-03$ | $3.40735e-02$ | $4.97992e-03$ | $1.43158e-02$ |
| 2nd order w/ CAAS | $6.87982e-03$ | $7.30759e-03$ | $3.40699e-02$ | $-7.18568e-05$ | $0.0$ |





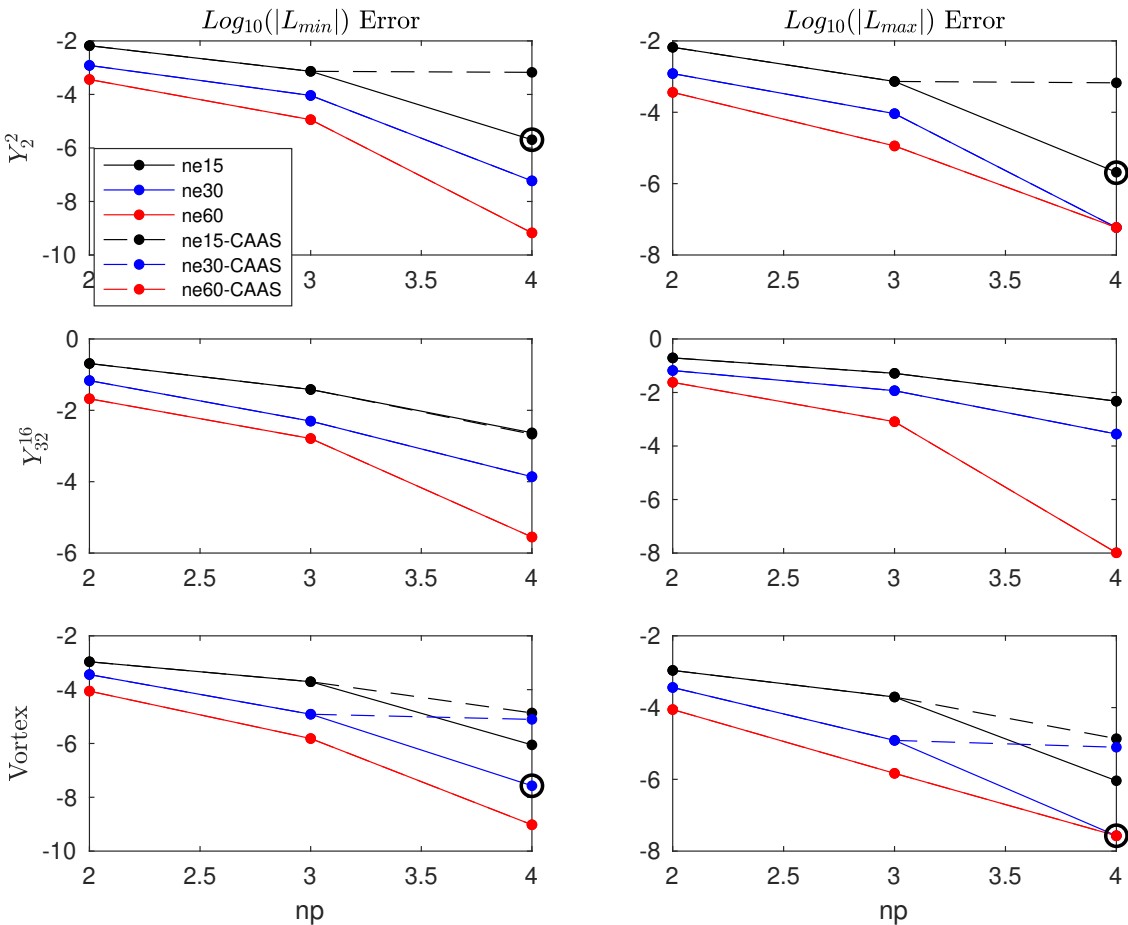

**Figure 4.** The $L_{max}$ and $L_{min}$ results of the convergence test for finite-element to finite-volume remapping from a cubed sphere to a latitude-longitude mesh using local bounds preservation for three different test cases. Circled data points indicate that the global minimum and maximum have been enhanced.

**Table 2.** Error Norms for remapping from an ne360 cubed sphere to a $0.25°$ degree latitude-longitude mesh.

| Method | $L_1$ | $L_2$ | $L_\infty$ | $L_{min}$ | $L_{max}$ |
|---|---|---|---|---|---|
| 1st order | $7.02963e-04$ | $1.16409e-03$ | $1.83186e-02$ | $-6.35094e-04$ | $0.0$ |
| 2nd order | $1.80731e-04$ | $2.34888e-04$ | $6.32974e-03$ | $-2.23843e-04$ | $4.68194e-03$ |
| 2nd order w/ CAAS | $1.80671e-04$ | $2.27702e-04$ | $6.32968e-03$ | $-2.30600e-04$ | $0.0$ |

## 4 Non-Conservative Monotone Remapping

In this section, we describe several different approaches to monotone remapping that are consistent but non-conservative. In

general, traditional approaches to monotone remapping perform poorly when the source mesh is significantly finer than the



**Figure 5.** Gibbs oscillations for a finite-volume to finite-volume remapping from a resolution 60 cubed sphere to a $1°$ latitude-longitude mesh, with 4th order polynomial reconstruction. Figure (a) shows the results without CAAS, figure (b) shows the results using CAAS with local bounds preservation, figure (c) shows the results using CAAS with local-p bounds preservation, and figure (d) shows the results using CAAS with global bounds preservation.

target mesh. To correct this, we propose what we call *integrated* approaches to remapping, which rely on the construction of the overlap mesh or supermesh (e.g., Farrell et al., 2009). This is in contrast to the more traditional approaches that amount to pointwise interpolations, which we refer to as *non-integrated* approaches, and are used extensively in, for instance, the Earth System Modeling Framework (Hill et al., 2004).

In brief, for consistent and monotone remapping operators, we express the value of $\psi^t$ at each spatial degree of freedom on the target mesh as a weighted sum of $N$ values of $\psi^s$:

$$\psi_j^t = w_{i_1}\psi_{i_1}^s + \ldots + w_{i_N}\psi_{i_N}^s, \tag{19}$$



**Figure 6.** One-dimensional cross sections for remapping of a discontinuous field at the equator. Figure (a) shows the results without CAAS, (b) shows the results using CAAS with local bounds preservation, (c) shows the results using CAAS with local-p bounds preservation, and (d) shows the results using CAAS with global bounds preservation.

where $i_k$ denotes the index of a spatial degree of freedom on the source mesh. As we're working with finite-volume meshes in this context, the spatial degrees of freedom correspond to the average value over the mesh faces. For consistency, we need $w_{i_1} + \ldots + w_{i_N} = 1$, and for monotonicity, we need $0 \leq w_{i_1}, \ldots, w_{i_N} \leq 1$. The weights $w_{i_k}$ then make up the entries of the remapping operator **R** given in equation (1).

## 4.1 Bilinear Remapping

Here, we describe the non-integrated approach to monotone bilinear remapping found in ESMF. Suppose we are given a point on the target mesh onto which we're remapping. Call this point $P_j$. First, we need to find a triangle or quadrilateral whose

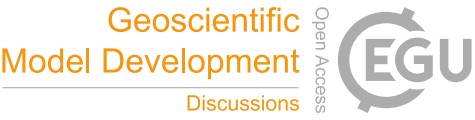

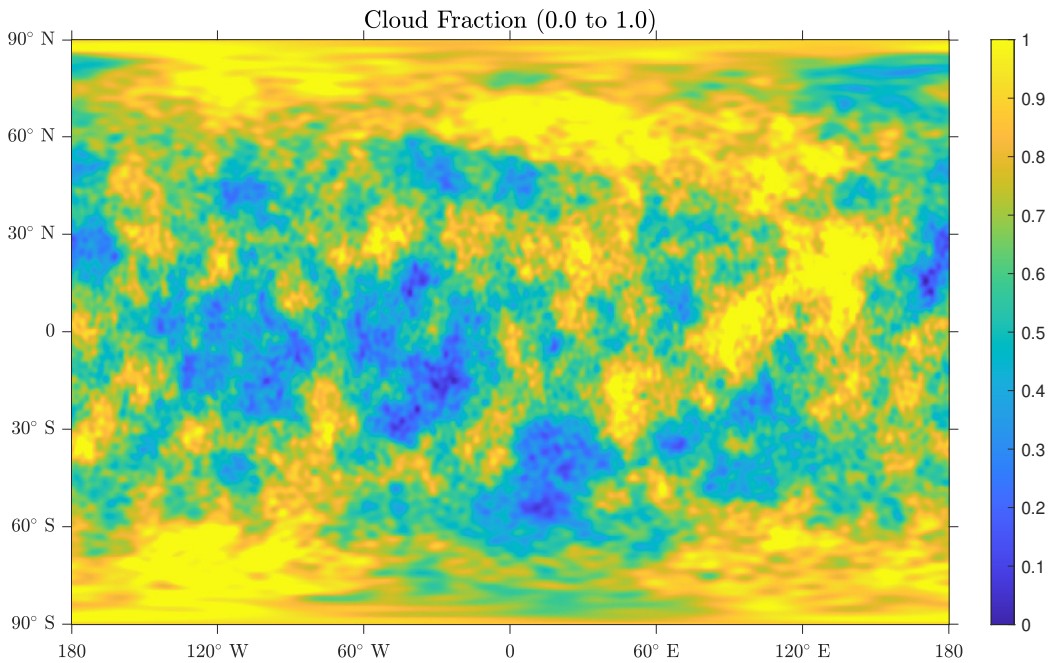

**Figure 7.** The cloud fraction field used in evaluating the effectiveness of CAAS on a real data field.

vertices are source face centers that contains $P_j$. We assume that the edges of these triangles and quadrilaterals are great-circle arcs. Now the source mesh is described in terms of the nodes of each face and the edges that connect them. Since the field values at the face centroids represent second-order approximations to the average value of the field, they are the natural choice for interpolation. To that end, the dual mesh of the source mesh is constructed, which will result in a mesh whose faces have source face centroids as their vertices. Once the dual mesh is available, it can be searched to find a polygon that contains $P_j$, the given point on the target mesh. If the polygon that contains $P_j$ has greater than four sides, it is triangulated in order to find a sub-triangle that contains $P_j$.

Once this polygon is found, and assuming it has to be further triangulated, we solve the following equation

$$(1 - \alpha - \beta)Q_{i_1} + \alpha Q_{i_2} + \beta Q_{i_3} = (1 - \gamma)P_j \tag{20}$$

where $Q_{i_1}$, $Q_{i_2}$, and $Q_{i_3}$ are the coordinates of the face centers of the triangle that contains $P_j$. Intuitively, the solution to this equation corresponds to first finding the intersection of the line through the origin and $P_j$, and the plane that passes through the points $Q_{i_1}$, $Q_{i_2}$, and $Q_{i_3}$, and then representing this point as a linear combination of these three points. The coefficients in





equation (20) then define the the value of the remapped field on the target mesh:

$$\psi_j^t = (1 - \alpha - \beta)\psi_{i_1}^s + \alpha\psi_{i_2}^s + \beta\psi_{i_3}^s. \tag{21}$$

Note that we have assumed that $P_j$ is the center of the $j$th face on the target mesh. The weights clearly sum to one, and they
are non-negative because triangle $Q_{i_1}Q_{i_2}Q_{i_3}$ contains $P_j$. Hence, this weighting defines a monotone, consistent remapping
operator (Ullrich and Taylor, 2015). The case where the polygon that contains $P_j$ has four sides is similar. In particular, we
solve the equation

$$(1-\alpha)(1-\beta)Q_{i_1} + \alpha(1-\beta)Q_{i_2} + \alpha\beta Q_{i_3} + \beta(1-\alpha)Q_{i_4} = (1-\gamma)P_j \tag{22}$$

where $Q_{i_1}, \ldots, Q_{i_4}$ are vertices of the quadrilateral that contains $P_j$. The coefficients in the previous equation then results in
the equation

$$\psi_j^t = (1-\alpha)(1-\beta)\psi_{i_1}^s + \alpha(1-\beta)\psi_{i_2}^s + \alpha\beta\psi_{i_3}^s + \beta(1-\alpha)\psi_{i_4}^s. \tag{23}$$

### 4.1.1 Delaunay Triangulation Remapping

In this section, we describe an alternative to the remapping scheme described in the previous section. We obviate the need
to triangulate an arbitrary polygon by constructing the Delaunay triangulation of the face centroids of the source mesh. We
outline our approach as follows. We seek a triangle on the source mesh whose vertices are source face centroids that contains
a given point on the target mesh. To that end, we divide the sphere into six panels, call them $R_i$ for $1 \leq i \leq 6$. The panels
$R_1, \ldots, R_4$ are equally sized and lie along the equator between $45°N$ and $45°S$. The panels $R_5$ and $R_6$ are equally sized caps
above and below $45°N$, and $45°S$, respectively. Let $S_i$ denote gnomonic projections of the set of source face centroids in $R_i$
onto the plane tangent to the sphere at the center of $R_i$. So $S_i$ is a set of two-dimensional points, and we denote its Delaunay
triangulation by $T(S_i)$. So given a point $P_j$ on the target mesh, we first find the panel $R_k$ that contains it. We then compute
$G(P_j)$, the gnomonic projection onto the plane tangent to the sphere at the center of $R_k$. We then find the triangle with vertices
$V_{i_1}, V_{i_2}, V_{i_3} \in T(S_k)$ that contains $G(P_j)$. Now we know that the gnomonic projection maps great circle arcs on the sphere to
straight lines on the plane, so if $P_{i_1}$, $P_{i_2}$, and $P_{i_3}$ are the points on the source mesh such that $G(P_{i_1}) = V_{i_1}$, $G(P_{i_2}) = V_{i_2}$, and
$G(P_{i_3}) = V_{i_3}$, then we can be sure that $P_j$ is contained within the spherical triangle whose vertices are $P_{i_1}$, $P_{i_2}$, and $P_{i_3}$. We
then approximate the value of $\psi_j^t$ as

$$\psi_j^t = \frac{A_{i_1}}{A}\psi_{i_1}^s + \frac{A_{i_2}}{A}\psi_{i_2}^s + \frac{A_{i_3}}{A}\psi_{i_3}^s \tag{24}$$

where, as can be seen in figure 8, $A_{i_k}$ is the area of the spherical subtriangle that does not have $P_{i_k}$ as a vertex, and $A$ is the
area of the spherical triangle that contains $P_j$. The weights are non-negative because they correspond to triangle areas, and they
are between zero and one because $0 \leq A_{i_k} \leq A$. Hence, these weights are monotone and consistent. Note that an advantage of
this approach is that it is easily parallelized, as we can divide the sphere into an arbitrary number of panels.


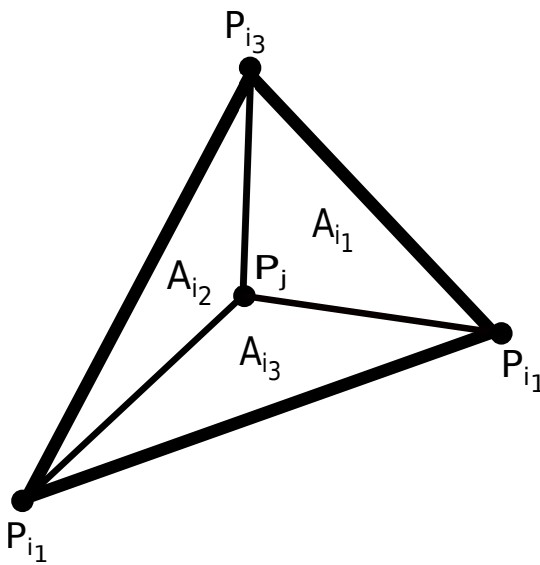

**Figure 8.** The weights used in the weighting based on the Delaunay triangulation.

### 4.1.2 Generalized Barycentric Coordinate Remapping

The final scheme we describe is based on what are called generalized barycentric coordinates (Floater, 2015). Our use of this scheme is motivated by a desire for a systematic way of incorporating more source points into equation (19). Intuitively, we expect such a scheme to give more accurate results, as it would incorporate more information from nearby source points for each point on the target mesh. We first define these coordinates, and then provide a description of where we expect them to be most useful. Let $P_1, \ldots, P_n$ be the vertices of a polygon in the plane, and $Q$ be a point within the polygon. The *generalized barycentric coordinates* of $Q$ with respect to the vertices, $w_i$, satisfy

1. $\displaystyle\sum_{i=1}^{n} w_i = 1$

2. $w_i \geq 0$

3. $\displaystyle\sum_{i=1}^{n} w_i(x) P_i = x$

The first two properties are responsible for consistency and monotonicity, and the third property, known as linear precision, means that linear functions can be reconstructed exactly in terms of the polygon vertices, and is essentially why these weights are second order accurate. One particular set of weights is given by the equation

$$w_i = A(P_{i-1}, P_i, P_{i+1}) \prod_{k \neq i, i-1} A(Q, P_k, P_{k+1}) \tag{25}$$





where $A(P_{i-1}, P_i, P_{i+1})$ and $A(Q, P_k, P_{k+1})$ denote the areas of triangles $P_{i-1}P_iP_{i+1}$, and $QP_kP_{k+1}$, respectively, and $w_i$ is the weight corresponding to vertex $P_i$ (Meyer et al., 2002).

   We generalize to the sphere by interpreting the areas in equation (25) as the areas of spherical triangles, rather than planar triangular areas. An advantage of these weights is that they are general; they can be used for arbitrary polygons, not just triangles and quadrilaterals.

As was the case with bilinear interpolation outlined in section 4.1, the dual of the source mesh is constructed. This will provide a mesh whose nodes are source face centers that can be searched through efficiently. We point out that for the triangular meshes we are considering, most faces on the dual mesh are hexagonal, so using the generalized barycentric coordinates given in equation (25) will allow up to six source points to be incorporated into the remapping operator in equation (19), instead of the three or four points that would be used for the Delaunay triangulation weighting given in equation (24), or the bilinear

weighting in equation (21). We hypothesize that this doubling of the amount of source points in equation (19) would lead to an increase in accuracy for remapping fields on triangular source meshes. Furthermore, the generalized barycentric coordinate weighting will always incorporate at least as many source points as either other scheme.

### 4.2   Non-Integrated Remapping: Numerical Tests

   Here we show the results of two different numerical tests. In the first case, the remapping is done from cubed spheres to a fixed

$1°$ latitude-longitude mesh. The cubed spheres are of increasing resolution with $ne = 5, 10, 20, 40, 80, 160$, and have $150, 600, 2400, 9600, 38400$, and $153600$ faces, respectively. The target mesh has $64800$ faces. We plot the error norms as functions of the approximate face size, which we take to be the square root of the approximate area of each face. Specifically, the face size is defined as

$$\text{face size} = \sqrt{\frac{4\pi}{N}}, \tag{26}$$

where $N$ is the number of faces. We see from figure 9 that the schemes converge at second order, and are approximately similar in magnitude.

   For the second test, the target is still a fixed $1°$ latitude-longitude mesh, and the source meshes are triangular geodesic meshes with $180, 720, 2880, 11520, 46080$, and $184320$ faces. From figure 10, we see that all schemes converge at second order and give similar error norms in most cases. For the high frequency and vortex tests, however, we see that the generalized barycentric

scheme gives consistently smaller $L_\infty$ errors than the Delaunay triangulation and the bilinear schemes, which indicates that the generalized barycentric weighting is slightly more effective at resolving the sharp gradients present in these fields. So although we hypothesized that the generalized barycentric coordinates would give perhaps a more noticeable improvement in the errors for all cases for triangular source meshes, we found that its benefit appears to be limited to the $L_\infty$ errors for the $Y_{32}^{16}$ and vortex fields.





**Figure 9.** Convergence results for several non-integrated monotone remapping schemes for a fixed latitude-longitude target mesh, and cubed sphere source meshes.





**Figure 10.** Convergence results for several non-integrated monotone remapping schemes for a fixed latitude-longitude target mesh, and triangular source meshes.



## 4.3 Integrated Remapping

The remapping schemes described in the previous sections work well when the source mesh is not too much finer than the target mesh. However, when the resolution of source mesh is greater than that of the target mesh, pointwise sampling of the source mesh to determine a field value on the target mesh is inappropriate and inaccurate. In this case, a large number of points on the source mesh contribute no weights to the remapping operator. To combat this under-sampling, we now describe an approach that ensures all points on the source mesh are sampled via construction of the overlap mesh or supermesh. Approaches of this type are called *integrated* because of their analogue to numerical quadrature, and are distinguished from the *non-integrated* approaches described in sections 4.1-4.1.2. A non-integrated approach basically amounts to an interpolation, whereby we express each value of the target field as a weighted sum of nearby source values. In the integrated approach, we recall that our variables correspond to face averages, and we approximate these integrals via quadrature. Specifically, for each face on the target mesh, we apply triangular quadrature to each sub-triangle of each overlap face, where the number of overlap faces is determined by the source mesh faces that intersect the given target face. Written out in full, we have

$$\psi_i^t \approx \frac{1}{|\Omega_i^t|} \sum_{j=1}^{N_{ov}} \sum_{k=1}^{N_T} \sum_{m=1}^{N_q} \psi^s(\boldsymbol{x}_{m,k,j}) dW_m \tag{27}$$

where $|\Omega_i^t|$ is the area of target face $i$, $N_{ov}$ is the number of source faces that overlap target face $i$, $N_T$ is the number of sub-triangles per overlap face, $N_q$ is the number of quadrature points per sub-triangle, $dW_m$ is the quadrature weight for $m$th quadrature point, and $\boldsymbol{x}_{m,k,j}$ is the location of the $m$th quadrature point within each sub-triangle of each overlap face. Now we don't know the value of $\psi^s(\boldsymbol{x}_{m,k,j})$, so we need to estimate it. In our numerical tests, we will use all of three of the weightings described in section 4. In particular, we estimate $\psi^s(\boldsymbol{x}_{m,k,j})$ as

$$\psi^s(\boldsymbol{x}_{m,k,j}) = \sum_{l=1}^{N} w_l \psi_{p_{i_l}}^t \tag{28}$$

where the $p_{i_1}, \ldots, p_{i_N}$, denote the faces on the source mesh whose centers form the polygon that contains $\boldsymbol{x}_{m,k,j}$, and $w_1, \ldots, w_{i_N}$ are the corresponding weights given by equations (24), (25), and (21) or (23), depending on the source mesh. Because the integration is performed by way of the overlap mesh, we can be sure that every degree of freedom on the source mesh contributes weights to the remapping operator.

### 4.4 Integrated Remapping: Numerical Tests

This section again consists of two tests. The first test is to establish second order convergence of the integrated schemes, and it is identical to the setup of the first test shown in section 4.2. In figure 11, we compare the results of the integrated versions of all three remapping schemes described in section 4. All three schemes give similar error norms to their non-integrated counterparts shown in figure 9. In particular, the error norms of the generalized barycentric, and bilinear schemes are nearly identical. This is to be expected, as in both schemes, the value at each point on the target mesh depends on four source points.



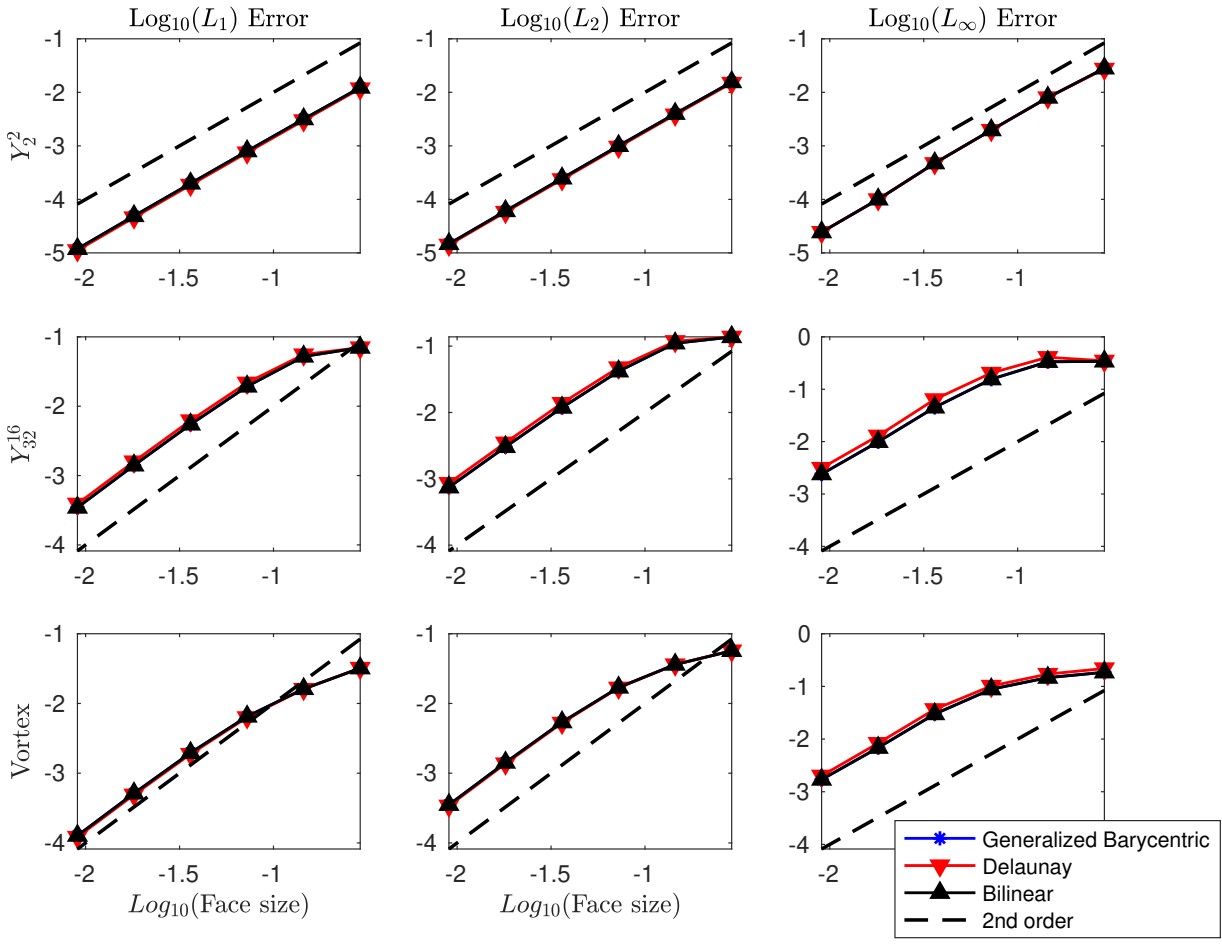

**Figure 11.** Convergence results for several integrated monotone remapping schemes for a fixed latitude-longitude target mesh, and cubed sphere source meshes.

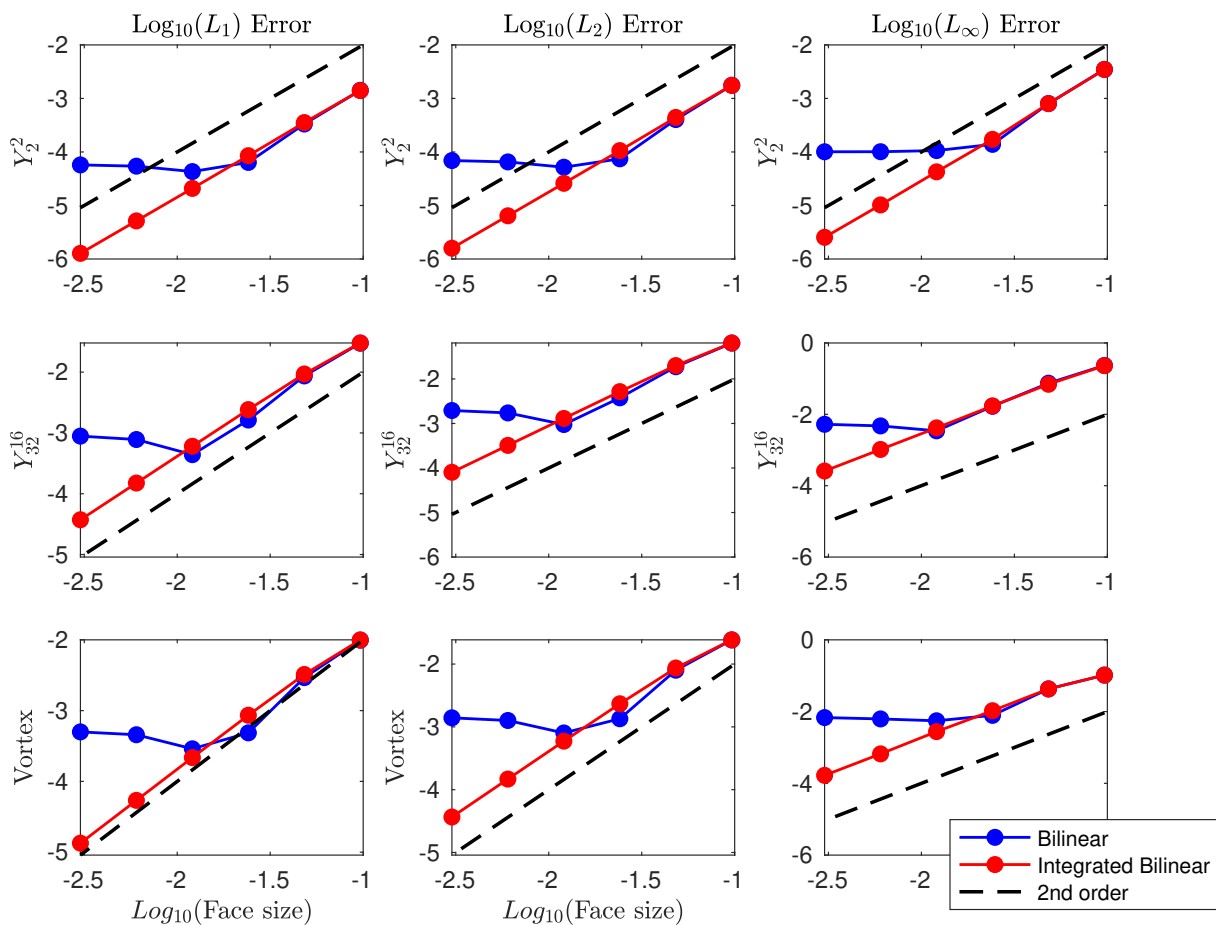

**Figure 12.** Convergence results for the integrated and non-integrated bilinear remapping schemes from cubed spheres to a fixed latitude-longitude mesh.

In the next test, we consider a setup where the source meshes are refined beyond the resolution of the target mesh. The source
meshes are cubed spheres with $ne = 15, 30, 60, 120, 240, 480$, and have $1350, 5400, 21600, 86400, 345600$, and $1382400$ faces, respectively. The target mesh is a fixed latitude-longitude mesh of $2°$ resolution. We see from figure 12 that the accuracy of the non-integrated scheme is degraded, but the integrated scheme remains second order. In particular, observe that the accuracy of the non-integrated bilinear scheme starts to diminish relative the the integrated one when the face size is between $10^{-1.5}$ and $10^{-2}$ which corresponds to a source mesh with no more than approximately $125000$ faces. Before this point, the errors of both
schemes are similar.



# 5 Conclusion

In this paper we have examined a number of different schemes for conservative and non-conservative monotone remapping. For monotone conservative remapping, we showed that the clip and assured sum method provides an accurate way of remapping conservative fields that are required to stay bounded, and is effective at reducing the Gibbs-like oscillations associated with
discontinuous source fields.

We then described several different approaches to non-conservative remapping. Two of these have, to the best of our knowledge, never been applied to remapping problems on the sphere. These methods have what are referred to as non-integrated and integrated versions, and it was shown that the integrated versions are capable of maintaining second accuracy across arbitrary source mesh resolutions by systematically sampling the degrees of freedom on the source mesh, albeit at higher computational
costs.

As discussed in the introduction, the methods described in this paper have been implemented as part of v2.1.6 of the TempestRemap software package (Ullrich et al., 2022).

*Code availability.* The code used in this paper is part of the Tempestremap software package, and is available on Zenodo (Ullrich et al., 2022).

*Author contributions.* DHM developed the functionality in tempestremap and wrote the paper, PAU advised and edited the manuscript

*Competing interests.* One of the (co-)authors is a member of the editorial board of Geoscientific Model Development. The peer-review process was guided by an independent editor, and the authors also have no other competing interests to declare.

*Acknowledgements.* Primary support for this work was provided by the SciDAC Coupling Approaches for Next Generation Architectures (CANGA) project, which is also funded by the DOE Office of Advanced Scientific Computing Research.



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
