# Peer review of "Strategies for Conservative and Non-Conservative Monotone Remapping on the Sphere"

_Geoscientific Model Development, 2022_

## Author Comment (AC1)

**Error for Vortex test case**

[Figure]

**Error for low frequency mode test case**

---

## Author Response (AR1)

**Author's Reply**

David H. Marsico and Paul A. Ullrich

The reviewers comments are highlighted in blue, and numbered with 1. The author's reply is in black, and numbered with 2. The changes in the manuscript are numbered with 3.

**Comments From Reviewer 1**

1. This paper provides a useful evaluation of remapping schemes between grids on the sphere, which is important for model intercomparison.

A few questions and comments:
1. "The largest errors occur when interpolating to the lat-long grid. Do we see particularly large errors at any point on the grid (e.g. poles or equator)."

2. In general, the errors are not the largest at the equator or poles for latitude-longitude meshes. They are largest in regions of high curvature, where we would expect there to be overshoots or undershoots.

1. "I think that it might also be relevant to mention conservative mass fixers developed for semi-Lagrangian schemes, such as the variants of Zerroukat, which make local corrections to fix both conservation and bounds."

2. The CAAS algorithm is similar to the one described in the paper "A simple mass conserving semi-Lagrangian scheme for transport problems" and we will include a mention of it in our revised manuscript.

3. The bibliography has been updated to include this reference, and we have referenced it in line 102.

1. "Please provide some more information about how R is constructed for each combination of grids."

2. For the conservative remapping, R is constructed according to a two stage procedure for finite-volume meshes. First, a fitting procedure is used to construct a local nth degree polynomial. This polynomial is then integrated over each overlap face that intersects a given target face (details can be found in the papers Arbitrary-Order Conservative and Consistent Remapping and a Theory of Linear Maps: Part I/II ). CAAS is then applied as a post-processing operation once R has been applied

to the source field.

For the non-conservative remapping, the entries of each row of R are determined by approximating each value on the target mesh as a weighted sum of values on the source mesh, as in equation (19). We consider both the non-integrated and integrated versions.

For the non-conservative non-integrated remapping, the non-zero entries of each row of R correspond to the faces on the source mesh whose centers form the nodes of a polygon that contain a given point on the target mesh. The values are then determined by whatever weights we're using, i.e. bilinear, generalized barycentric, or Delaunay triangulation interpolation. Details can be found in lines 182-189.

For the non-conservative integrated remapping, R is constructed by way of the overlap mesh. By using the overlap mesh, we ensure that every face on the source mesh is sampled. Specifically, we approximate the value of the field on each target mesh face by integrating the source field over that target face by applying numerical quadrature to each intersecting overlap face (see equations (27) and (28) for this written out fully). A description of this construction is included in section 4.3.

3. We have added a reference of how R is constructed in the conservative case near line 106.

1. "I'm confused by the reference of supermeshing in the nonconservative section - this technique is introduced usually to ensure conservation."

2. While the overlap mesh is generally used to ensure conservation, we have adapted it to be used in a non-conservative context. For non-conservative remapping, the overlap mesh provides a systematic way of ensuring that every point on the source mesh is sampled, because the overlap mesh consists of all faces common to both the source and target meshes. The sample points used are obtained by triangulating the super mesh faces.

**Comments From Reviewer 2**

This was an interesting paper and I thought that overall the ideas were good and potentially useful. (There are definitely some things in this paper that I'm interested in trying in our code.) However, I thought that some of the sections could be expanded a bit to make the overall algorithm clearer (e.g. 4.1.2). The one thing that I saw that should be changed is under "Requested minor revisions" below. The comments below that in the "Questions and comments" section are just suggestions.

1. Requested minor revisions: The one thing that I saw that should probably be changed is that you make a general statement about how it was shown that the integrated versions are capable of maintaining accuracy across arbitrary source mesh resolutions (line 318). However, in section 4.4 test 2 you only show the bilinear

results for integrated vs. non-integrated for a source refined beyond the resolution of the target mesh. Given this, I think that you should either add graphs for the other 2 non-conservative methods in that second test or just mention the bilinear in that conclusion sentence. It could be that I'm misunderstanding what's being shown in that section (4.4.), if so a bit more explanation in there about why just bilinear is being shown in the second test would be useful. Also, I think "arbitrary" is a bit strong for that sentence, maybe something like "wide range" would be better to describe what you show.

2. All of the integrated versions of the schemes are capable of maintaining second order accuracy when the source mesh resolution is refined significantly beyond that of the target mesh, not just the bilinear one. Thank you for pointing out a potential source of confusion, and we'll add a clarifying sentence. The reason why we only included bilinear is because the figures look nearly identical in all cases, and we thought it would sufficient to just show one, and then mention how the other schemes are similar.

3. We have added an explanation as to why we only included the results for the bilinear integrated schemes at the end of section 4.4, and changed the word "arbitrary" to "a wide range of."

1. Questions and comments:

1. - Line 26: I wondered if you meant "non-conservative" at the end of this line, since you talk about conservative in the next part.

2. We do mean conservative here. The next sentence, the one that begins with "In the conservative case. . .", is meant to describe the way monotonicity in the conservative case has been achieved in other contexts, i.e. through limiters. In the sentence after that, we describe the specific way we achieve monotonicity, i.e. through the CAAS algorithm.

1. Line 191: ESMF also supports regridding where the data values are on the nodes, so dual conversion isn't always necessary.

2. Thank you for sharing this useful information with us.

1. - Section 4.1.1. It would be useful to have a diagram showing how this algorithm works (e.g. with the 6 panels on the sphere showing a coarse triangulation and destination point.)

2. Thank you for this advice. We hope that the new figure we have added will be useful.

3. We have included a new figure (figure 8). The figure shows a simple representation of how the source faces on each panel are projected onto a plane and then

triangulated. We also reference this figure in line 226.

1.     - Line 220: What happens if a set of source point spans two panels? (e.g. do you have an overlap region so that a destination point can't land between two panels)

2. Yes, we have a such a region of about 10 degrees on each side of the panel to prevent something like this.

1.     - Line 221: You could add a sentence or two about how you find the triangle that contains the point (e.g. do you just loop or is there a search structure involved)

2. We use a kd-tree to accomplish this. First, we use the tree to find the triangle whose center is nearest to the target point. If the target point is in this triangle, we stop. Otherwise we search through neighboring faces until we find one that contains the target point. This algorithm works efficiently in practice.

3. We have added description of this process at line 224.

1.     - Section 4.1.2: I thought that the broader algorithm could be fleshed out a bit more so that the description was at a similar level to other sections. Even a few sentences describing how you find the polygon that would contain the point (or a pointer if you're doing it the same as in another section)

2. Thank you for the suggestion. We have added more of a description of this algorithm and how it is similar to the other ones in use.

3. We have added a description of this at line 256.

1.     - Line 245: Does this scheme for calculating the weights work if the polygon is concave?

2. In general, the scheme will not necessarily work for concave faces. However, it can be adapted by triangulating it and then applying the weighting formula for each sub-triangle.

1.     - Line 318 it says " second accuracy"  should it be  "second order accuracy"?

2. Thank you for pointing out this typo.

3. We have changed it to "second order accuracy."

**Comments From Reviewer 3**

1. This study was an inspiring and interesting paper on remapping. We wanted to experiment with some of the ideas from this paper on our system. In the process, I thought it would be nice if there were additional explanations to help understand

them.

2. Thank you for these comments.

1. Many readers of this paper, including myself, will desire to compare which remapping method is the best from a monotonicity. The picture needs to be improved to help them understand intuitively. For example, it is proposed to unify the axis (y-axis in Figures 1-4, 9-11, and 12) and color range (Figures 5 and 7) for comparison between figures. In addition, in order to distinguish the dense lines in Fig. 9-11, more distinct markers should be used or a table containing actual error values should be included (suggested to add as a supplementary material).

2. Thank you for these suggestions. We have changed the axes and color ranges in these figures which we describe below. Figure 7 is unrelated to the plots in figure 5. It shows the global cloud fraction, not the idealized vortex test case of figure 5.

3. In figures 1-4 and figures 10-13, each subplot has the same set of axes. We have also changed the marker size in figures 10-13 so that it is easier to see. In figure 5, the subplots now have the same color range.

1. It seems hard to aware that the color scheme in figure 5a is not bounded on purpose. It would be nice if there was a mention of this in the text or caption. The modified figure reflects the reviewer's intention well.

2. In order to have a consistent color scheme for all four subplots, we used the maximum and minimum values of the remapped field obtained without using CAAS to define the upper and lower bounds.

3. The updated figures with a consistent set of color schemes are given in figure 5.

1. In the remapping result of the vortex case (Fig. 5), I would like to comment on the reason why a very conspicuous irregular pattern occurs around (0, 90 N). In particular, I wonder why these errors are weakened (Fig 5b) when a local bound is applied.

2. Yes, there are significant errors in the remapped field when CAAS is not applied. This is because high-order methods lead to overshoots and undershoots of the global bounds. Applying CAAS with local bounds preservation will prevent these overshoots/undershoots because the remapped fields are constrained to like between 0 and 1.

1. I'm still wondering why the patterns that are removed from the CAAS local bound (fig 5b) are not removed when using the local-p bound (fig 5c).

2. CAAS with local bounds enforces a much tighter form of bound preservation than CAAS with local-p bounds, which is why the oscillatory patterns appear in 5c but not 5b.

1. It could be out of purpose of this study, but it is curious about efficiency, another desirable property of the remapping operator. I wonder how long each of the remapping methods used in this study takes to calculate. Also, if possible, I would like to hear answers about whether there is a dependency on the data-type of variable.

2. The non-integrated schemes are significantly faster than their integrated counterparts. This is because the non-integrated schemes do not rely on the overlap mesh, but the integrated versions do. The overlap mesh can be very large, and it therefore takes much longer to iterate over.

It took approximately half an hour to generate the offline maps for the integrated schemes for a cubed sphere source mesh with 1,382,400 faces and for a target mesh with 16,200 faces. It never takes more than a few minutes to generate the maps for the non-integrated schemes. There does not seem to be a dependency on the data-type.

1. - Additional minor typos.

L51: In the second part we, we show » In the second part, we show

L61: a a set of discrete nodes » a set of ...

2. Thank you for pointing these out.

3. We have fixed the typos.